# Smooth Regularization for Efficient Video Recognition

**Gil Goldman**
Computer Science Department
Carnegie Mellon University
`gilg@andrew.cmu.edu`

**Raja Giryes**
School of Electrical and Computer Engineering
Tel-Aviv University
`raja@tauex.tau.ac.il`

**Mahadev Satyanarayanan**
Computer Science Department
Carnegie Mellon University
`satya@cs.cmu.edu`

## Abstract

We propose a smooth regularization technique that instills a strong temporal inductive bias in video recognition models, particularly benefiting lightweight architectures. Our method encourages smoothness in the intermediate-layer embeddings of consecutive frames by modeling their changes as a Gaussian Random Walk (GRW). This penalizes abrupt representational shifts, thereby promoting low-acceleration solutions that better align with the natural temporal coherence inherent in videos. By leveraging this enforced smoothness, lightweight models can more effectively capture complex temporal dynamics. Applied to such models, our technique yields a $3.8\%$–$6.4\%$ accuracy improvement on Kinetics-600. Notably, the MoViNets model family trained with our smooth regularization improves the current state-of-the-art by $3.8\%$–$6.1\%$ within their respective FLOP constraints, while MobileNetV3 and the MoViNets-Stream family achieve gains of $4.9\%$–$6.4\%$ over prior state-of-the-art models with comparable memory footprints. Our code and models are available at `https://github.com/cmusatyalab/grw-smoothing`.

## 1 Introduction

Video recognition has rapidly evolved over the past decade, with models becoming increasingly capable of learning sophisticated spatial and temporal representations. However, despite remarkable advancements, many architectures still suffer from overfitting or inefficient use of temporal information [7, 12, 28]. In light of these challenges, we introduce a novel smooth regularization approach designed to instill a strong inductive bias specifically tailored to video data. The key insight behind our method is that video content often exhibits continuous motion and gradual changes in appearance, suggesting that representations should vary smoothly over time. By explicitly encouraging this smoothness, we aim to guide neural networks toward more stable and generalizable internal feature representations, ultimately leading to improved performance across a range of video recognition tasks.

Our regularization strategy focuses on the intermediate-layer embeddings produced by a neural network when processing consecutive frames. Instead of allowing these embeddings to fluctuate arbitrarily across frames, we constrain their dynamics to resemble a Brownian motion, which translates to imposing Gaussian Random Walk (GRW) behavior in the frames discrete settings that promotes continuous and relatively modest rates of change. The inspiration behind this modeling choice comes from the fact that, in a typical video, adjacent frames exhibit only gradual shifts in object positioning, scale, lighting, or motion. By treating frame-to-frame representation shifts as a form of GRW, we incorporate a principled, mathematically grounded way to preserve smoothness in

39th Conference on Neural Information Processing Systems (NeurIPS 2025).

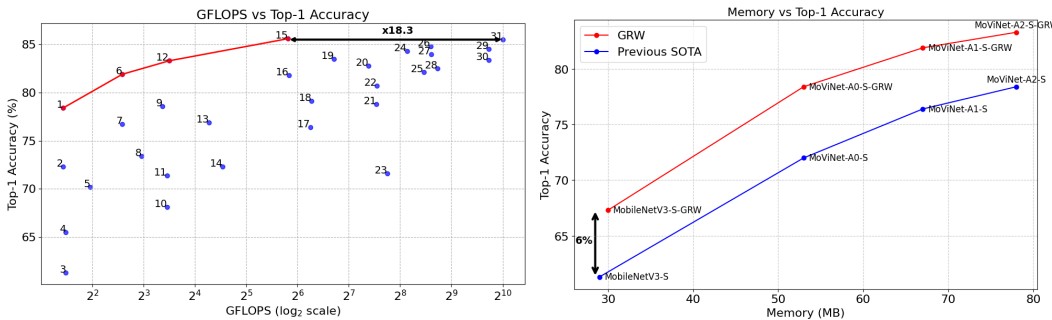

Figure 1: **Performance Results on Kinetics-600.** By simply adding GRW-smoothing to existing models, we achieve significant improvements. **Left:** Accuracy vs. FLOPs, where each point corresponds to a published model (see Table 1 for references). GRW-smoothing improves the state-of-the-art performance of efficient models by 3.8–6.1%. Notably, MoViNet-A3-GRW achieves 85.6% accuracy at just 56.4 GFLOPs, while the closest model, MViTv2-B-32×3, requires 18.3× more FLOPs. **Right:** Accuracy vs. Memory. GRW-smoothing improves the state-of-the-art performance of memory-efficient models by 4.9–6.4%.

the learned embeddings. This not only reflects a more natural representation of videos but also acts as a regularizing force against abrupt or erratic changes in the network's internal states.

A core outcome of our approach is that it naturally discourages large jumps between successive embeddings. Mathematically, our formulation involves adding the GRW penalty term to the training objective, which grows whenever the model produces excessively rapid transitions between frames in the network embedding space. By penalizing these abrupt shifts, we encourage networks to learn features that evolve more gently over time. This favors solutions that maintain a sense of temporal consistency, that is, low acceleration in the embedding space, leading to more coherent internal representations that can better capture the true temporal dynamics present in real-world videos.

By aligning model training with the natural temporal structure found in the data, our approach makes it easier to networks to focus on learning meaningful temporal correlations rather than wasting capacity on fitting noisy or abrupt changes. Consequently, the model becomes more sensitive to subtle motion cues, often crucial for recognizing fine-grained actions or micro-movement semantics without sacrificing robustness to variations due to noise in the network embedding space.

While bigger network may have enough capacity to learn both the variations and noise in the embedding space together with the changes in motion, this is more challenging in resource constrained networks. Thus, we focus in this work on such type of networks. We demonstrate the effectiveness of our proposed smooth regularization on such lightweight models. To verify its benefits, we trained these smaller-scale networks on the popular Kinetics-600 dataset, a large benchmark known for its diversity of human action classes. By simply adding our novel loss function to the training of these architectures, we get consistent gains of 3.8%–6.4% in classification accuracy as shown in Figure 1, leading to new state-of-the-art performance under FLOP and memory constraints.

Our main contributions can be summarized as follows:

**Smoothness Prior in Video Recognition**: We introduce a novel regularization technique that enforces smoothness in the intermediate-layer embeddings of consecutive video frames by modeling their changes as a GRW.

**State-of-the-Art Performance Under Efficiency Constraints**: Our technique outperforms current leading solutions within a similar memory and compute range, confirming the broad applicability of our method to resource-limited scenarios.

**Flexible Framework**: Focusing on smoothness as a strong inductive bias provides a plug-and-play regularization option for existing video recognition pipelines. It integrates seamlessly into different architectures with a negligible computational overhead as illustrated in Figure 3.

## 2   Related Work

**Video recognition in general.**   Early approaches to video action recognition learned spatiotemporal representations with 3D CNNs [5, 27], with subsequent advances such as Two-Stream networks [26] and SlowFast [9] improving the balance between spatial semantics and motion modeling. Temporal segment sampling [30] provided an efficient way to cover long videos with sparse clips. With the advent of transformers, attention-based models extended self-attention to spatiotemporal tokens [2, 1]. Subsequent work introduced hierarchical and multiscale designs and localized attention to improve efficiency [6, 22], and explored alternative attention mechanisms such as trajectory attention [24], as well as MLP-like backbones [32] and video-specific ViT adaptations [25]. Despite strong accuracy, pure transformers often incur substantial computational and memory costs for long videos, particularly relative to compact CNN baselines [2, 1, 22].

**Lightweight video recognition.**   A parallel line of work targets real-time and on-device deployment by reducing compute and parameters. *CNN-based* designs dominate this space due to depthwise separable convolutions and efficient temporal operators. MoViNets [15] introduce a NAS-designed family of efficient 3D CNNs and a stream buffer for constant-memory streaming, achieving strong accuracy–efficiency trade-offs. X3D [8] systematically compounds network width, depth, and temporal resolution to yield compact 3D CNNs. Temporal Shift Module (TSM) [21] augments 2D CNN backbones with a zero-parameter, zero-FLOP temporal exchange, enabling video recognition with image-classification-level compute. Further lightweight temporal modules include TEA [19], TDN [29], and TAdaConv [13], which inject temporal cues with modest overhead. *Transformer-based* models such as MViT [6] and VideoSwin-T [22] reduce attention cost via multiscale hierarchies or windowed attention, yet they often remain heavier than the most compact CNN baselines at strict mobile budgets. *Hybrid* architectures combine convolutions and attention to leverage local inductive bias with global modeling. UniFormer [18] exemplifies this trend by interleaving convolutional blocks and self-attention. Overall, lightweight video recognition in practice is led by CNNs and convolution–attention hybrids (see Table 1 in the Experiments section below).

**Temporal coherence: slowness and higher-order smoothness.**   An early prominent approach to temporal coherence, slow feature analysis (SFA) [31] prefers features that evolve as slowly as possible in time by minimizing the expected squared temporal derivative of each feature subject to zero mean, unit variance, and decorrelation constraints. The motivation is that latent factors in natural videos typically change gradually, so maximally slow features capture stable, semantically meaningful structure.

**Temporal order and ranking constraints.**   A complementary line of self-supervised work focuses on verifying or predicting the chronological order of frames or clips. Representative approaches include Shuffle & Learn [23], Sorting Sequences (Order Prediction Networks) [16], and Odd-One-Out networks [10]. These objectives encourage representations to respect temporal structure by reasoning about sequence order rather than enforcing slowness.

## 3   Method

Consider a video frame sequence $X = (\mathbf{x}_t)_{t=0}^{M-1}$ and an encoding of a video recognition model's intermediate layer $\varphi(X) = Z = (\mathbf{z}_t)_{t=0}^{N-1}$, *where $M$ and $N$ denote the numbers of input frames and embedding time steps (after any temporal subsampling), respectively.* The main objective of this work is to guide the optimization process to favor solutions $\varphi$ for which $\mathbf{z}(t)$ is a smooth function of $t$.

**Warm-up Example.** Let us consider an instructive simplified example. We constructed a small dataset containing 1,000 short videos of a few model airplanes performing one of three rotations: Yaw, Pitch, or Roll, starting from a random initial position, as shown in Figure 2(top).

To analyze the geometry of the embeddings, we trained two identical models. In both, we use a pretrained MobileNet as the recognition model that calculates embeddings per frame and then a single Transformer layer that process several consecutive frames for the temporal information. The models are trained to predict the rotation label using cross-entropy loss. In the second model, we smooth

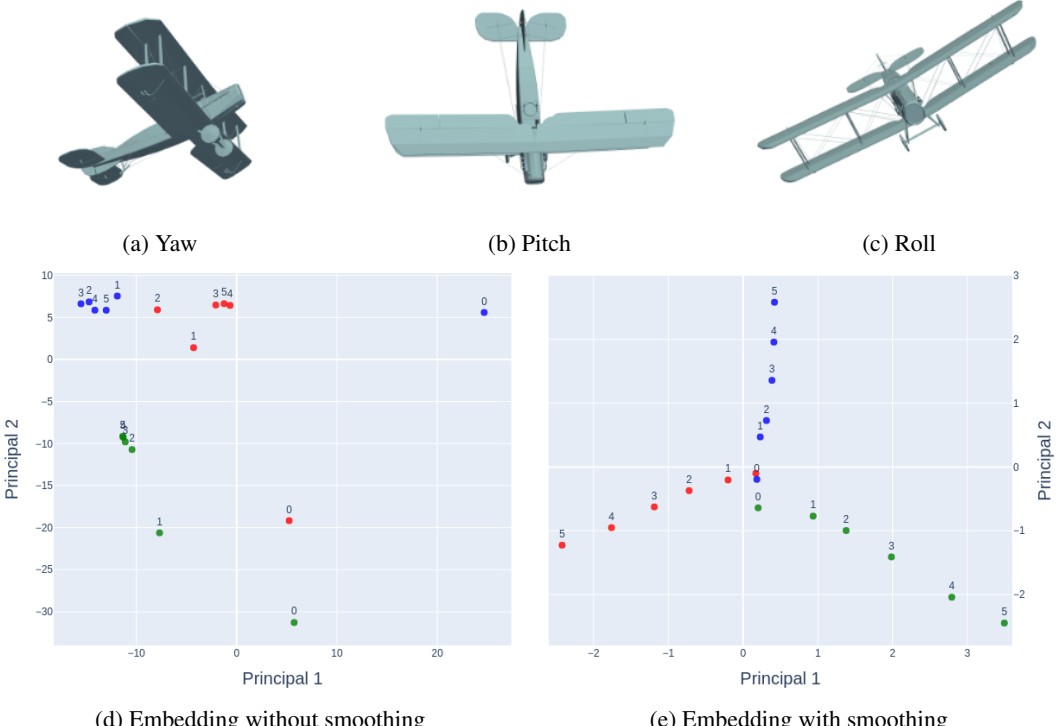

Figure 2: **Warm-up Example.** *Top*: The used Airplanes dataset containing 1,000 training and 100 test short videos of model airplanes performing one of three rotations, starting from a random position. The dataset isolates temporal classification, as any single frame is independent of the rotation label. *Bottom*: Output embeddings of two identical models trained with and without the smoothness term. In green, blue and red are typical clips embeddings for Yaw, Pitch and Roll, respectively, projected to the first two principal components of the embedded test set. Each point is a single frame embedding. The index is the clip frame index.

the MobileNet embeddings $Z$ with an additional loss term that penalizes high accelerations in $\mathbf{z}(t)$, directing the optimization towards keeping $\frac{d^2}{dt^2}\mathbf{z}(t)$ low. *This term, which is the main contribution of the paper,* is formally described in Section 3.1. The resulting embeddings without and with this smoothing term are presented in Figures 2d and 2e respectively.

The geometry in Figure 2d shows that the standard encoder lacks a smooth structure. As it does not utilize the smooth prior we have with respect to video sequences, a more complicated function was learned than simply the movement in yaw/pitch/roll. In contrast, Figure 2e shows that the model trained with the smoothness term finds an intrinsic linear two-dimensional representation where each rotation is mapped to a certain direction. Note that the curve in the plot shows that each rotation is smooth, and the accelerations $\frac{d^2}{dt^2}\mathbf{z}(t)$ are low.

We next turn to explain how we calculate smoothing term that we suggest for a given normalized embedding in a neural network.

## 3.1   The Gaussian Random Walk (GRW) Smoothing Term

Consider a normalized layer output $Z = (\mathbf{z}_t)_{t=0}^{N-1}$ over time. Our goal is to induce a smoothness prior on the embeddings. We do it within a $T$ time window, dividing $Z$ into short subsequences:

$$Z^c = (\mathbf{z}_0^c, \dots, \mathbf{z}_{T-1}^c) \coloneqq (\mathbf{z}_{cT}, \dots, \mathbf{z}_{(c+1)T-1}), \quad c = 0, \dots, C-1, \ \ C = \lfloor N/T \rfloor. \tag{1}$$

Imposing a direct smooth prior on $Z$ poses a difficulty, as mapping all $\mathbf{z}(t)$ to a single point is "maximally smooth" but results in a degenerate solution that is clearly undesired. Therefore, we construct the smooth loss in two steps.

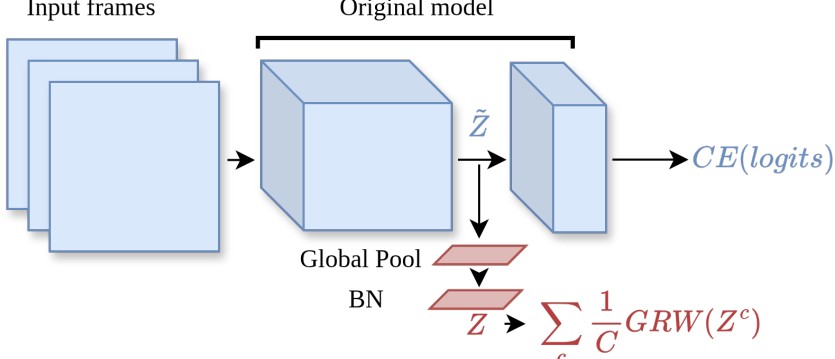

Figure 3: **Intermediate layer smoothing.** The encodings $\tilde{Z}$ are global-pooled along the spatial dimensions, then normalized across the batch dimension, where we use BN without learnable parameters. The sub-clips $Z^c$ are fed into GRW.

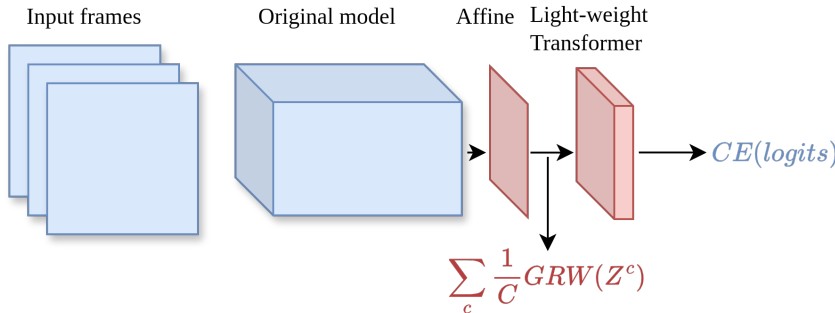

Figure 4: **Final layer smoothing.** Output encodings $\varphi(X) = \tilde{Z}$ of a given video model are affine transformed to $Z$. The sub-clips $Z^c = (\mathbf{z}_{cT}, \ldots, \mathbf{z}_{(c+1)T-1})$ are fed into GRW regularization, as an additional loss term, then further processed using a few Attention layers.

1. *Frame Ordering:* We first introduce a contrastive loss that directs the optimization towards mappings that maintain the structure of the order of the frames.

2. *Smooth Prior:* We then impose a distribution that favors low-acceleration mappings and plug it in the contrastive loss, resulting in our smoothing prior.

**Frame Ordering.** Consider the following *right-frame-order* contrastive loss

$$\mathcal{L}_f(\varphi) = - \mathop{\mathbb{E}}_{X,c} \left[ \log \frac{f(\mathbf{z}_0^c, \mathbf{z}_1^c, \mathbf{z}_2^c, ..., \mathbf{z}_{T-1}^c)}{\sum_{\pi \in S(1:T)} f(\mathbf{z}_0^c, \mathbf{z}_{\pi(1)}^c, \mathbf{z}_{\pi(2)}^c, ..., \mathbf{z}_{\pi(T-1)}^c)} \right], \qquad (2)$$

where $f$ is a probability distribution that we will define in the next step of the smooth prior, and $S(1 : T)$ is the group of all permutations $\pi$ of the elements $\{1, \ldots, T - 1\}$. That is, we fix the first frame and contrast the correct ordering of the remaining frames with all their permutations. This prevents the loss term from degenerate solutions that collapse to the same point.

**Smooth Prior.** In the loss term in Equation (2), $f$ can be chosen freely. We impose the smoothness prior by setting $f$ as a distribution favoring low acceleration embedding.

Define the velocities and accelerations of the embedding

$$\frac{d}{dt} Z^c = V^c = (\mathbf{v}_t^c)_{t=0}^{T-2} := (\mathbf{z}_1^c - \mathbf{z}_0^c, ..., \mathbf{z}_{T-1}^c - \mathbf{z}_{T-2}^c),$$

$$\frac{d}{dt} V^c = A^c = (\mathbf{a}_t^c)_{t=0}^{T-3} := (\mathbf{v}_1^c - \mathbf{v}_0^c, ..., \mathbf{v}_{T-2}^c - \mathbf{v}_{T-3}^c).$$

To smooth $\mathbf{z}(t)$ we model the distribution of the velocities as random walk with Gaussian increments,

$$\mathbf{v}_t^c | \mathbf{v}_0^c = \mathbf{v}_0^c + \sum_{i=0}^{t-1} \mathbf{a}_i^c, \qquad\qquad t = 1, \ldots, T-2, \qquad\qquad (3)$$

where $(\mathbf{a}_t^c)_{t=0}^{T-3}$ are i.i.d., $\mathbf{a}_t^c \sim \mathcal{N}(\mathbf{0}, I)$. Under this assumption

$$f(Z^c) := p(\mathbf{v}_1^c, \ldots, \mathbf{v}_{T-2}^c | \mathbf{v}_0^c) = p(A^c) = \prod_{t=0}^{T-3} \mathcal{N}(\mathbf{a}_t^c), \qquad\qquad (4)$$

where, with abuse of notation, $\mathcal{N}$ denotes the density of the standard normal distribution. To put it all together, the loss in Equation (2) becomes

$$\mathcal{L}(\varphi) = - \mathop{\mathbb{E}}_{X,c} \left[ \log \frac{p(A^c)}{\sum_{\pi \in S(1:T)} p(A_\pi^c)} \right], \qquad\qquad (5)$$

where $A_\pi^c$ are the accelerations according to the permutation $\pi$, $A_\pi^c := \frac{d^2}{dt^2}(\mathbf{z}_0^c, \mathbf{z}_{\pi(1)}^c, \mathbf{z}_{\pi(2)}^c, \ldots, \mathbf{z}_{\pi(T-1)}^c)$.

The loss (5) requires a scaling parameter. For any embedding $\varphi$ the scaling of the embedding by $\alpha$ to $\alpha\varphi$ introduce an inverse temperature parameter

$$\mathcal{L}(\alpha\varphi) = - \mathop{\mathbb{E}}_{X,c} \left[ \log \frac{p(\alpha A^c)}{\sum_{\pi \in S(1:T)} p(\alpha A_\pi^c)} \right].$$

We determine the scaling by adding another term $\Omega(V^c)$ controlling the unconditional speeds

$$\mathbf{v}_t^c \sim \mathcal{N}(\mathbf{0}, I), \qquad \Omega(V^c) = \log \prod_{t=0}^{T-2} \mathcal{N}(\mathbf{v}_t^c).$$

The final smooth prior is

$$\mathcal{L}_{smooth}(\varphi) = - \mathop{\mathbb{E}}_{X,c} \left[ \log \frac{p(A^c)}{\sum_{\pi \in S(1:T)} p(A_\pi^c)} + \alpha\Omega(V^c) \right], \qquad\qquad (6)$$

and the final loss is

$$\mathcal{L}_{CE} + \lambda L_{smooth}. \qquad\qquad (7)$$

Note that the sum in Equation 6 is taken over all permutations, which grow factorially with $T$. For large $T$, we uniformly sample $k$ permutations, for some given $k$. In practice, we enumerate all $(T-1)!$ orderings when $T \le 7$ and uniformly sample $k = 1000$ permutations when $T > 7$. Since $(T-1)! < 1000$ for $T \le 7$, the number of evaluated orderings per clip is $\le 1000$, keeping the computational cost of the denominator in Equation (6) effectively independent of $T$.

### 3.2 Applying GRW to a neural network

We propose to apply the GRW loss to intermediate, typically higher, layers of video recognition models to induce the smoothness inductive bias. In Section 4 we demonstrate empirically the advantage of using this loss term.

We propose applying the GRW term in two possible locations in a neural network: (i) smoothing of an intermediate layer, or (ii) smoothing of the final layer. We describe each option next.

**Intermediate Layer Smoothing (Figure 3).** For an intermediate layer output $\tilde{Z} \in \mathbb{R}^{C \times N \times K}$, where $N$ is the temporal dimension, $C$ is the channel dimension, and $K$ is the (flattened) spatial dimension, the encodings are globally pooled along the spatial dimensions and then normalized to have expected value of $\mathbf{0} \in \mathbb{R}^C$ and mean unit length,

$$Z = \frac{1}{\sqrt{C}} BN_{1d}(GP(\tilde{Z})) \in \mathbb{R}^{C \times N},$$

where the $BN_{1d}$ does not have learnable shift and rescale parameters. Then, we extract sub-clips $Z^c$ from $Z$ and use them as input to the GRW loss.

**Final Layer Smoothing (Figure 4).** We smooth the final encodings of the model, just before the classification head and then further process the smoothed embeddings with a lightweight temporal model, namely, a Transformer with 1-2 layers. Formally, the final layer output $\tilde{Z}$ is normalized using a learnable affine transformation, $Z = Linear(\tilde{Z})$, which is applied on each embedding separately. In the supplementary material we show that optimizing this transformation together with the GRW loss, put the embeddings in $Z$ close to the unit sphere and therefore we refer to this affine transformation as normalization. Also here, we create from $Z$ the sub-clips $Z^c$ and use them in the GRW loss.

## 4    Experiments

To demonstrate the effect of smoothing using our method, we compare the performance of models trained with and without GRW regularization.

**Datasets.** We report our results on Kinetics-600 (K600) [4] and Kinetics-400 (K400) [14]. Both datasets consist of 10-second videos of varying resolutions and frame rates, labeled with 600 and 400 action classes.

**Models and Implementation.** We focus on video recognition models with lower computational requirements to control training time, and since introducing inductive bias becomes increasingly important when efficiency is a factor. We selected the current state-of-the-art models in the lowest categories of FLOPS and memory, and fine-tuned them by applying GRW-smoothing. Specifically, we applied GRW to the MoViNet model family A0,...,A3, and their streaming versions A0-S,...,A2-S. The streaming variants, denoted as Ai-S, are memory-efficient versions of the MoViNet-Ai models that process videos frame by frame. The baseline performance of these streaming models is lower than that of their non-streaming counterparts due to the use of causal operations.

At the lowest end of memory requirements, we applied GRW to MobileNetV3 Small with the following modification. We extract a "base frame" every $T$ frames, where $T$ is the GRW clip window (see Section 3). All frames within a $T$-frame window are processed individually, but alongside their corresponding base frame. To support this, we modified the first layer to accept 6 input channels.

We used Final Layer GRW-smoothing (see Figure 4), which produced better results than Intermediate Layer GRW-smoothing (Figure 3). However, both methods improve accuracy results, as discussed in Section 4.2. We maintain the original model's final layer output dimension in the normalization affine transformation and apply GRW-smoothing on the output. We replace the classification head with a 2-layer vanilla transformer with a standard $\times 4$ MLP expansion factor (see Section 3 for details).

We set $\lambda = 10^{-1}$ as the balancing factor in Equation (7) and $\alpha = \frac{1}{2}$ as the scaling factor in Equation (6). We found that the results are robust to perturbing the values of $\lambda$, which suggests that gradients in the direction of smooth solutions align with gradients with respect to the classification likelihood; see ablation studies in Subsection 4.2.

In all experiments we set the GRW window to span $0.5$–$1.0\,$s of video. Specifically, for MoViNet-A0/-A1/-A2-GRW and MobileNetV3-S-GRW we use $5\,$fps with $T{=}5$ (covering $1\,$s), and for MoViNet-A3-GRW we use $12\,$fps with $T{=}6$ (covering $0.5\,$s). For these values of $T$ we enumerate the full set of orderings in Equation (6), of size $(T{-}1)!$ (i.e., 24 for $T{=}5$ and 120 for $T{=}6$), so no permutation subsampling was required (i.e., $k$ was not used).

**Remark 1.** *GRW-smoothing operates on per-frame embeddings $Z = (\mathbf{z}_t)$ rather than raw frames, so the added computation is small relative to the backbone. For example, in MoViNet-A2-S-GRW the input frame $\mathbf{x}_t \in \mathbb{R}^{224 \times 224 \times 3}$ (150,528 dims) is mapped to $z_t \in \mathbb{R}^{640}$, i.e., an $\approx 235 \times$ reduction in dimension. Consequently, the $O(k \cdot T)$ vector operations and log-probability evaluations in Equation (6) are negligible compared to the early convolutional blocks. Empirically, with the $T$ values used in this section (full enumeration of permutations), the wall-clock time per training epoch for MoViNet-A2-S-GRW is only 2% higher than MoViNet-A2-S without smoothing; even for larger $T$ where one might sample up to $k{=}1000$ permutations, the overall cost remains dominated by the backbone forward/backward pass.*

**Training.** On K600, we fine-tune starting from existing weights. On K400, we use transfer learning from K600. We employ a simple training process, not applying augmentations except when training

the A2 and A3 models on K400. We use different training rates for the transformer head and model backbone, decreasing with a cosine learning rate scheduler in the range $[10^{-4}, 10^{-6}]$ for the model backbone and $[10^{-3}, 10^{-5}]$ for the transformer head. We fine-tune for 14 epochs on K600 and 10 epochs on K400.

The smaller models, A0–A1 and MobileNet, were trained on a single `dgx-A100` for 3–5 days, while the A2 and A3 models were trained on $2 \times$ `dgx-A100` for 5 days.

## 4.1 Results

For all results, we use a single clip evaluation for our models. We report Top-1 accuracy results against FLOPS and memory. The resolution column refers to the resolution of the input video, with 224 indicating $224 \times 224$. The frames column is given by $num\_clips \times num\_frames$ used in the evaluation. The GFLOPs column indicates the *total* computation for the evaluation of a single video sample.

As seen in Table 1, all models with GRW-smoothing achieve significant improvements in accuracy and set new SOTA results in their corresponding GFLOPs group. Specifically, MoViNet-A0-S-GRW, MoViNet-A1-S-GRW, MoViNet-A2-S-GRW and MoViNet-A3-GRW improve the SOTA results by 6.1%, 5.2%, 4.7% and 3.8%, respectively. For MoViNet-A3-GRW, the next model achieving similar accuracy, MViTv2-B-32×3, requires $18.3\times$ more GFLOPs.

We compare current SOTA memory-efficient models, namely MobileNet and streaming versions of the MoViNet model family, before and after smoothing them with GRW; see Table 2. MobileNetV3-S-GRW, MoViNet-A0-S-GRW, MoViNet-A1-S-GRW, MoViNet-A2-S-GRW improve their non-smooth versions by 6.0%, 6.4%, 5.5% and 4.9%, respectively; see also Figure 1(right).

## 4.2 Ablation Studies

We study the effect of GRW by (i) disentangling the contribution of smoothing versus the added attention layers, and (ii) analyzing sensitivity to the key hyperparameters: the GRW window $T$, the scaling factor $\alpha$ in Equation (6), and the balance $\lambda$ in Equation (7).

**Placement and attention vs. smoothing.** We ablate where GRW is applied (Intermediate vs. Final layer) and whether the attention head alone explains the gains. Concretely, on K600 we train: (i) MoViNet-A2-S-GRW (Final Layer Smoothing; Figure 4), (ii) MoViNet-A2-S-GRW (Intermediate Layer Smoothing; Figure 3), and (iii) MoViNet-A2-S + attention, the baseline equipped with the same 2-layer Transformer head but *without* GRW. Table 4 shows that adding attention alone yields a small gain over the baseline (+0.9 Top-1), while training with GRW yields an additional +4.0 absolute (for a total +4.9 over the baseline) when applied at the final layer; applying GRW at an intermediate layer also improves accuracy (+2.4) even without attention.

**Sensitivity to $T$, $\alpha$, and $\lambda$.** We further ablate the GRW hyperparameters on MoViNet-A0-S-GRW trained on K600, varying one parameter at a time and keeping all other settings fixed (Final Layer Smoothing; training protocol as in Sec. 4). Results are summarized in Table 5.

**Summary.** Across these ablations, GRW is not overly sensitive near the settings used in our main results: $T=5$, $\alpha=\frac{1}{2}$ in Equation (6), and $\lambda=10^{-1}$ in Equation (7). Short windows under-exploit temporal coherence, very long windows may blur distinct motions, and excessively large $\lambda$ can over-regularize.

## 5 Conclusion

In summary, we introduced a novel smooth regularization technique designed to enhance temporal understanding in video recognition models, particularly lightweight architectures. By modeling the evolution of frame embeddings as a Gaussian Random Walk, our method penalizes abrupt representational changes, effectively promoting low-acceleration solutions that align with the natural temporal coherence of video data. This approach has demonstrated significant accuracy improvements, notably a 3.8%–6.4% gain on Kinetics-600, and has established new state-of-the-art performance in compute-constrained settings. By combining our proposed GRW regularization with models such as

Table 1: K600 by FLOPS

| | Model | Top-1 | GFLOPs | RES | FRAMES |
|---|---|---|---|---|---|
| 1 | **MoViNet-A0-S-GRW** | **78.4** | 2.7 | 172 | $1 \times 50$ |
| 2 | MoViNet-A0[15] | 72.3 | 2.7 | 172 | $1 \times 50$ |
| 3 | MobileNetV3-S[11] | 61.3 | 2.8 | 224 | $1 \times 50$ |
| 4 | MobileNetV3-S+TSM[21] | 65.5 | 2.8 | 224 | $1 \times 50$ |
| 5 | X3D-XS[8] | 70.2 | 3.9 | 182 | $1 \times 20$ |
| 6 | **MoViNet-A1-S-GRW** | **81.9** | 6.0 | 172 | $1 \times 50$ |
| 7 | MoViNet-A1[15] | 76.7 | 6.0 | 172 | $1 \times 50$ |
| 8 | X3D-S[8] | 73.4 | 7.8 | 182 | $1 \times 40$ |
| 9 | MoViNet-A2[15] | 78.6 | 10.3 | 224 | $1 \times 50$ |
| 10 | MobileNetV3-L[11] | 68.1 | 11.0 | 224 | $1 \times 50$ |
| 11 | MobileNetV3-L+TSM[21] | 71.4 | 11.0 | 224 | $1 \times 50$ |
| 12 | **MoViNet-A2-S-GRW** | **83.3** | 11.3 | 224 | $1 \times 50$ |
| 13 | X3D-M[8] | 76.9 | 19.4 | 256 | $1 \times 50$ |
| 14 | X3D-XS[8] | 72.3 | 23.3 | 182 | $30 \times 4$ |
| 15 | **MoViNet-A3-GRW** | **85.6** | 56.4 | 256 | $1 \times 120$ |
| 16 | MoViNet-A3[15] | 81.8 | 56.9 | 256 | $1 \times 120$ |
| 17 | X3D-S[8] | 76.4 | 76.1 | 182 | $30 \times 13$ |
| 18 | X3D-L[8] | 79.1 | 77.5 | 356 | $1 \times 50$ |
| 19 | MoViNet-A4[15] | 83.5 | 105 | 290 | $1 \times 80$ |
| 20 | UniFormer-S[18] | 82.8 | 167 | 224 | $4 \times 16$ |
| 21 | X3D-M[8] | 78.8 | 186 | 256 | $30 \times 16$ |
| 22 | X3D-L[8] | 80.7 | 187 | 356 | $1 \times 120$ |
| 23 | I3D[4] | 71.6 | 216 | 224 | $1 \times 250$ |
| 24 | MoViNet-A5[15] | 84.3 | 281 | 320 | $1 \times 120$ |
| 25 | MViT-B-16×4[6] | 82.1 | 353 | 224 | $5 \times 16$ |
| 26 | MoViNet-A6[15] | 84.8 | 386 | 320 | $1 \times 120$ |
| 27 | UniFormer-B[18] | 84.0 | 389 | 224 | $4 \times 16$ |
| 28 | XViT (8×)[3] | 82.5 | 425 | 224 | $3 \times 8$ |
| 29 | XViT (16×)[3] | 84.5 | 850 | 224 | $3 \times 16$ |
| 30 | MViT-B-32×3[6] | 83.4 | 850 | 224 | $5 \times 32$ |
| 31 | MViTv2-B-32×3[20] | 85.5 | 1030 | 224 | $5 \times 32$ |

**Table 1:** *Top-1 accuracy, total video evaluation cost (in GFLOPs), input resolution (RES), and FRAMES = clips × frames per clip used for evaluation on Kinetics-600. Models enhanced with our proposed smooth regularization are marked with GRW. These models consistently outperform their baselines and other state-of-the-art methods under similar FLOP constraints. Variance: for MoViNet-A0-S-GRW, across three seeds we obtain $78.4 \pm 0.05$ Top-1 (mean ± std).*

MoViNet-A0/1/2/3 and their streaming counterparts, as well as MobileNetV3, we improve overall performance within their respective FLOP and memory constraints. The GRW regularization acts as a flexible, plug-and-play component with minimal computational overhead, guiding networks towards more stable and generalizable feature representations.

While our approach presents promising results, it has certain limitations. The core assumption of Gaussian Random Walk dynamics, while beneficial for many natural videos, might not be universally optimal for content characterized by extremely abrupt transitions or intentionally discontinuous motion. A future work may explore such videos and how to extend GRW for such cases. Furthermore, while our experiments demonstrate significant gains on lightweight models, the extent of improvement on very large-capacity models, which we could not do due to computational constraints, require further investigation. Finally, the necessity of the frame ordering component in the contrastive loss, while effective in preventing degenerate solutions, does introduce an additional layer of complexity to the training objective. A more efficient variants can be studied in a future work.

Looking ahead, several avenues for future research emerge. Extending the application of our GRW smoothing to a wider array of video architectures, including more complex Transformer-based models, could yield further insights into its generalizability. Investigating its efficacy across diverse video understanding tasks beyond action recognition, such as temporal action localization or video

Table 2: K600 by Mem

| Model | Top-1 | Mem MB |
|---|---|---|
| MobileNetV3-S[11] | 61.3 | 29 |
| **MobileNetV3-S-GRW** | **67.3** | 30 |
| **MoViNet-A0-S-GRW** | **78.4** | 53 |
| MoViNet-A0-S[15] | 72.0 | 53 |
| **MoViNet-A1-S-GRW** | **81.9** | 67 |
| MoViNet-A1-S[15] | 76.4 | 67 |
| **MoViNet-A2-S-GRW** | **83.3** | 78 |
| MoViNet-A2-S[15] | 78.4 | 78 |

**Table 2:** *Top-1 accuracy and memory usage (in MB) for memory-efficient models on Kinetics-600. Models enhanced with our smooth regularization (GRW) are shown in bold and consistently outperform their baselines under identical memory constraints.*

Table 3: K400 by FLOPS

| Model | Top-1 | GFLOPs |
|---|---|---|
| **MoViNet-A0-S-GRW** | **70.4** | 2.7 |
| MoViNet-A0[15] | 65.8 | 2.7 |
| MoViNet-A2[15] | 75.0 | 10.3 |
| **MoViNet-A2-GRW** | **77.6** | 11.3 |
| X3D-XS[8] | 69.5 | 23.3 |
| **MoViNet-A3-GRW** | **81.7** | 56.4 |
| MoViNet-A3[15] | 78.2 | 56.9 |
| X3D-S[8] | 73.5 | 76.1 |
| VideoMamba[17] | 76.9 | 108 |

**Table 3:** *Top-1 accuracy, total video evaluation cost, on Kinetics-400.*

Table 4: Ablation on placement (K600, MoViNet-A2-S family).

| Model | Top-1 | GFLOPs |
|---|---|---|
| **MoViNet-A2-S-GRW (final layer)** | **83.3** | 11.3 |
| MoViNet-A2-S-GRW (intermediate layer) | 80.8 | 10.3 |
| MoViNet-A2-S + attention (no GRW) | 79.3 | 11.3 |
| MoViNet-A2-S[15] | 78.4 | 10.3 |

anomaly detection, presents another promising direction. Additionally, a dynamic smoothing window that adapts to video content is favorable. Finally, a more in-depth theoretical understanding of how GRW regularization influences the optimization landscape and feature learning process would be beneficial.

# 6   Acknowledgements

Parts of this research were conducted using ORCHARD, a high-performance cloud computing cluster. The authors would like to acknowledge Carnegie Mellon University for making this resource available to its community.

This material is based upon work supported by the United States Navy under award number N00174-23-1-0001 and by the National Science Foundation under grant number CNS-2106862. The content of the information does not necessarily reflect the position or the policy of the government and no official endorsement should be inferred. This work was done in the CMU Living Edge Lab, which is supported by Intel, Arm, Vodafone, Deutsche Telekom, CableLab, Crown Castle, InterDigital, Seagate, Microsoft, the VMware University Research Fund, IAI, and the Conklin Kistler family fund. Any opinions, findings, conclusions or recommendations expressed in this document are those of the authors and do not necessarily reflect the view(s) of their employers or the above funding sources.

This work was partially supported by a grant from The Center for AI and Data Science at Tel Aviv University (TAD).

| (a) Window $T$ | | (b) $\alpha$ in Equation (6) | | (c) $\lambda$ in Equation (7) | |
|---|---|---|---|---|---|
| **Top-1 (%)** | $T$ | **Top-1 (%)** | $\alpha$ | **Top-1 (%)** | $\lambda$ |
| 77.3 | 3 | 77.9 | 0.25 | 78.0 | 0.01 |
| **78.4** | **5** | **78.4** | **0.5** | **78.4** | **0.1** |
| 78.0 | 10 | 78.3 | 1.0 | 75.5 | 1.0 |
| 72.0 | no smoothing | 72.0 | no smoothing | 72.0 | no smoothing |

Table 5: Hyperparameter ablations for GRW on K600 with MoViNet-A0-S-GRW. (a) Window $T$ peaks at $T=5$. (b) Scaling $\alpha$ shows a mild optimum near $\alpha=0.5$ in Equation (6). (c) Balance $\lambda$ is robust in $\{0.01, 0.1\}$ and degrades at $\lambda=1.0$ in Equation (7).

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

# A  Scaling

Here we show the scaling behavior of GRW-smoothing. The main result we prove is that the optimal solution for a smoothing window of size $T$, applied to approximately centered data, lies within a ball of radius bounded by $\mathcal{O}(T\sqrt{\ln T})$. We prove this result in the one-dimensional case.

Let us recall the setup:

Consider $T \geq 3$ points in $[0, R]$ with fixed endpoints

$$0 =: z_1 \leq z_2 \leq \cdots \leq z_T := R, \qquad Z = (z_t)_{t=1}^T, \tag{8}$$

and define velocity and acceleration vectors as

$$V(Z) = (z_2 - z_1, \ldots, z_T - z_{T-1}) \in \mathbb{R}^{T-1}, \qquad v = (v_t)_{t=1}^{T-1},$$

$$A(Z) = (v_2 - v_1, \ldots, v_{T-1} - v_{T-2}) \in \mathbb{R}^{T-2}, \qquad A = (a_t)_{t=1}^{T-2}.$$

We will denote by $\mathcal{Z}^T \subset \mathbb{R}^T$ the set of all point configurations of the form (8), that is, non-decreasing sequences with $z_1 = 0$. For any $Z \in \mathcal{Z}^T$ we denote $R(Z) := z_T$.

For any such configuration $Z$, define the loss components:

$$\mathcal{L}_v(Z) = \frac{1}{2} \sum_{t=1}^{T-1} v_t^2 = \frac{1}{2} \sum_{t=1}^{T-1} (z_{t+1} - z_t)^2,$$

and

$$\mathcal{L}_a(Z) = -\log \frac{\exp\left(-\frac{1}{2} \sum_{t=1}^{T-2} a_t^2(Z)\right)}{\sum_\pi \exp\left(-\frac{1}{2} \sum_{t=1}^{T-2} a_t^2(Z^\pi)\right)},$$

where the sum is over all permutations $\pi$ of $\{2, \ldots, T\}$ fixing $z_1$, and

$$Z^\pi = (z_1, z_{\pi(2)}, \ldots, z_{\pi(T)}).$$

With the above notation and $\alpha = 1$ (see Equation (6) in the paper), GRW loss is given by

$$\mathcal{L}(Z) = \mathcal{L}_a(Z) + \mathcal{L}_v(Z).$$

**Theorem 1** (*GRW*-smoothing scale). *Given* $T \geq 3$, *let* $Z^* = \arg\min_{Z \in \mathcal{Z}^T} \mathcal{L}(Z)$. *Then*

$$R(Z^*) = \mathcal{O}(T\sqrt{\ln T}).$$

**Remark 2.** *Theorem 1 suggests that for approximately centered embedding, it is beneficial to allow a scaling degree of freedom before feeding embedding vectors into GRW-smoothing to allow the solution to converge into the $\mathcal{O}(T\sqrt{\ln T})$ ball. Therefore, when applying GRW-smoothing to a sequence model embedding, we either directly scale and center the embedding using a non-learnable batch normalization (BN), or allow for a learnable linear transformation to set the scaling. Empirically, we found that centering is not required when a linear transformation is applied.*

*Proof.* We will use the following propositions in the proof, providing their proofs subsequently:

**Proposition 1** (Uniform Lower Bound on $\mathcal{L}$). *For any* $Z \in \mathcal{Z}^T$,

$$\mathcal{L}(Z) \geq \mathcal{L}_v(Z) \geq \frac{R^2(Z)}{2(T-1)}. \tag{9}$$

**Proposition 2** (Uniform Configuration Upper Bound). *Consider the uniform configuration*

$$Z_u = \left(0, \frac{R}{T-1}, \frac{2R}{T-1}, \ldots, R\right). \tag{10}$$

*For* $R = T - 1$,

$$\mathcal{L}(Z_u) = \mathcal{O}(T \ln T). \tag{11}$$

Assuming Proposition 1 and Proposition 2, we will now complete the proof of Theorem 1 and provide their proofs subsequently.

From Proposition 1 and Proposition 2, it follows that

$$\frac{R^2(Z^*)}{2(T-1)} \leq \mathcal{L}(Z^*) \leq C_1 T \ln T,$$

for some constant $C_1$. Rearranging, we get the desired result.

Now we prove the auxiliary claims.

*Proof of Proposition 1.* Since $\mathcal{L}_a(Z)$ is non-negative, we have $\mathcal{L}(Z) \geq \mathcal{L}_v(Z)$. Recall that $\mathcal{L}_v(Z) = \frac{1}{2} \sum_{t=1}^{T-1} v_t^2$. For any $Z$ the velocities are non-negative and satisfy $\sum_t v_t = z_T - z_1 = R(Z) := R$. Therefore, $\mathcal{L}_v(Z) \geq \min_{V \in \mathbb{R}^{T-1}, V \geq 0, \|V\|_1 = R} \frac{1}{2} \|V\|_2^2$. The last is a classic quadratic program with the minimizer $V_u = \frac{R}{T-1}(1, ..., 1)$, attaining the minimum $\frac{R^2}{2(T-1)}$, where these velocities are realized by the uniform configuration of the points. Hence, we obtain $\mathcal{L}(Z) \geq \mathcal{L}_v(Z) \geq \frac{R^2(Z)}{2(T-1)}$. $\square$

*Proof of Proposition 2.* Fix $R = T - 1$ and consider the uniform configuration (10). We have

$$\mathcal{L}(Z_u) = \underbrace{\ln \left( \sum_{\pi:\pi(1)=1} \exp\left( -\frac{R^2}{2(T-1)^2} S(\pi) \right) \right)}_{\mathcal{L}_a(Z_u)} + \underbrace{\frac{R^2}{2(T-1)}}_{\mathcal{L}_v(Z_u)},$$

where

$$S(\pi) := \sum_{t=1}^{T-2} (\pi(t+2) - 2\pi(t+1) + \pi(t))^2.$$

The velocity term simplifies as

$$\mathcal{L}_v(z_u) = \frac{R^2}{2(T-1)} = \frac{(T-1)^2}{2(T-1)} = \frac{T-1}{2}. \tag{12}$$

The acceleration term simplifies as

$$\mathcal{L}_a(z_u) = \ln \left( \sum_{\pi:\pi(1)=1} \exp\left( -\frac{1}{2} S(\pi) \right) \right) \leq \ln((T-1)!) = \mathcal{O}(T \ln T). \tag{13}$$
$\square$

Then by (12) and (13) $\mathcal{L}(z_u) = \mathcal{O}(T \ln T)$.

$\square$

