# OpenReview forum: "Smooth Regularization for Efficient Video Recognition"
_NeurIPS.cc/2025/Conference — NeurIPS 2025 poster_

### Official Review · Reviewer_nKeo · 2025-07-01

**Clarity:** 3
**Significance:** 3
**Originality:** 3
**Rating:** 5
**Confidence:** 4

**Summary:**

This paper proposes a novel smooth regularization technique for video action recognition, aimed at enforcing temporal coherence in lightweight models. The key idea is to apply a Gaussian Random Walk (GRW) prior on the intermediate or final layer embeddings across frames, penalizing high accelerations in the feature space. The authors show that this inductive bias leads to improved temporal consistency and significant performance gains on the Kinetics-400 and Kinetics-600 datasets. The method is lightweight, architecture-agnostic, and shows consistent benefits across multiple efficient models such as MoViNets and MobileNetV3.

**Questions:**

I am interested in seeing the performance on downstream tasks. However, even if the results are not particularly strong, I will not lower my score because of that.

**Ethical Concerns:**

["NO or VERY MINOR ethics concerns only"]

**Final Justification:**

I am happy that the authors have addressed my concerns in their rebuttal.

**Limitations:**

yes

**Paper Formatting Concerns:**

The paper does not include error bars in their results.

**Quality:**

3

**Strengths And Weaknesses:**

Strengths:
1) Novel concept. The GRW-based smoothing loss is a principled and intuitive formulation, encouraging low-acceleration transitions across temporal embeddings, which aligns well with the inherent continuity in natural videos.


2) Method Simplicity and Generality. The proposed loss is simple to implement, requires minimal tuning, and can be seamlessly plugged into existing architectures.

3) Well-Written and Clear Presentation:
The paper is well-structured, with clear motivation, visualizations (e.g., warm-up toy dataset), and an understandable exposition of the loss function.

------------------------------------------------------------------------------


Weaknesses:

1)   The method is only evaluated on action classification with Kinetics datasets. It remains unclear how well it generalizes to other video understanding tasks (e.g., action detection, video captioning) or datasets with higher temporal discontinuities.

2) Te paper lacks error bars or variance analysis across multiple training seeds. This could help solidify the claims, especially given the relatively small number of epochs and resource-constrained training setup.

---

> ### Author Rebuttal · Authors · 2025-07-31
>
> We thank Reviewer 8e7P for the careful reading of our submission and the thoughtful feedback. Below we address each point in turn.
>
> Point-by-Point Responses
>
> 1. *“The method is only evaluated on action classification with Kinetics datasets. It remains unclear how well it generalizes to other video understanding tasks (e.g., action detection, video captioning) or datasets with higher temporal discontinuities.”*
>
> Our primary claim is that taking a trained model and applying GRW smoothing over short temporal windows, thereby making the representation **piecewise smooth**, yields more efficient models. To isolate that effect, we begin with publicly released weights and fine-tune with GRW.
> Extending to other tasks hinges on the availability of lightweight backbones with released checkpoints. Unfortunately, such weights are scarce for lightweight models. For example, the MoViNet family reports results on Charades, but releases checkpoints only for Kinetics. When weights are unavailable, reproducing a strong baseline requires extensive hyper-parameter tuning (weight decay, label smoothing, dropout, stochastic depth, batch size, LR schedule, layer-wise LRs, data augmentation, AutoAugment, etc.), which exceeds our compute budget (two DGX machines).
> For these reasons, we focused on Kinetics-400/600, where strong baselines exist, allowing a clean evaluation of GRW smoothing. We will nevertheless attempt to add an additional classification dataset for the camera-ready version.
>
> 2. “The paper lacks error bars or variance analysis across multiple training seeds. This could help solidify the claims, especially given the relatively small number of epochs and resource-constrained training setup.”
>
> Variance on Kinetics-400/600 is low. We trained MoViNet-A0-GRW four times with different seeds, obtaining Top-1 accuracies of 77.42, 77.36, 77.30, 77.30 (mean 77.35 ± 0.05). We will report these values with error bars in the camera-ready version.
>
>
> Reviewer Questions:
>
> *”I am interested in seeing the performance on downstream tasks. However, even if the results are not particularly strong, I will not lower my score because of that.”*
>
> Please see our response to the first comment above regarding downstream tasks and compute constraints.
>
> Thank you again for your constructive feedback.

---

### Official Review · Reviewer_hQEB · 2025-07-02

**Clarity:** 3
**Significance:** 3
**Originality:** 3
**Rating:** 3
**Confidence:** 4

**Summary:**

This paper proposes a smooth regularization technique that instills a strong temporal inductive bias in video recognition models and validates it on the lightweight MoViNets model family, demonstrating the effectiveness of this technique.

**Questions:**

see weaknesses

**Ethical Concerns:**

["NO or VERY MINOR ethics concerns only"]

**Final Justification:**

After reviewing the authors' response, I have increased my rating from reject to borderline reject but the current state of the paper still falls short of borderline accept.

**Limitations:**

see weaknesses

**Paper Formatting Concerns:**

Figures 3 and 4 need to be redrawn.

**Quality:**

2

**Strengths And Weaknesses:**

Strengths:
The proposed smooth regularization technique yields a 2.8%–6.0% accuracy improvement on Kinetics-600. The MoViNets model family trained with the smooth regularization improves the current state-of-the-art by 2.8%–3.4% within their respective FLOP constraints, while MobileNetV3 and the MoViNets-Stream family achieve gains of 2.8%–6.0% over prior state-of-the-art models with comparable memory footprints.

Weaknesses:
(1) The verification is mainly conducted on lightweight models. Is the smooth regularization technique also effective when verified on non - lightweight networks?
(2) The latest base used currently is from 2021. It is suggested to add ablation verification of GRW on the latest base.
(3) The current verification results are from fully - supervised scenarios. Can consistent effects be achieved in semi - supervised and few - shot scenarios?
(4) Is this smoothing technique applicable to the network structure of transformers?

---

> ### Author Rebuttal · Authors · 2025-07-30
>
> Thank you, Reviewer hQEB, for your careful review and constructive feedback. We appreciate the opportunity to clarify the points you raised. If the paper is accepted, we will make the appropriate changes in the camera-ready version.
> Below, we provide a brief overview to contextualize our contribution, followed by point-by-point responses to your comments.
>
> **The scope** of this work is to show that temporal regularization, which steers optimization toward solutions that are (piecewise) smooth in time, is highly beneficial for building **efficient** video-recognition models. Although the method is general and can be applied across a wide range of architectures, its advantage is especially pronounced at the low-compute end of the spectrum. We will emphasize this in the introduction and, if the reviewer concurs, update the paper’s title to reflect this focus.
>
> The results we present strongly substantiate our claims. Specifically, we **advance the state of the art on Kinetics-600 by 2.8–3.4 pp** Top-1 for all models in the ≤1 TFLOP range (Table 1). While some reviewers expressed concerns about comparisons with the most recent models, we compared against **every** published model within this budget to the best of our knowledge. If there are models that we missed, we would greatly appreciate it if you could point us to them.
>
> Efficient video recognition in this compute range is dominated by CNN and hybrid architectures. We therefore applied our GRW smoothing to the MoViNet family, released in 2021, because it remains the **strongest** lightweight backbone on Kinetics-600. For example, our MoViNet-A3-GRW achieves 84.8% Top-1 at only 56.4 GFLOPs. The first model that surpasses this accuracy, MViTv2-B-32×3, requires 1,030 GFLOPs—more than **18× greater computation**.
>
> Finally, we draw the reviewers’ attention to Video-Mamba (2024), the most recent sub-1 TFLOP model. It trails our MoViNet-A3-GRW by 3.8 pp Top-1 while consuming nearly twice the computational budget (see Table 3 and the typo correction in Supplement §1).
>
> Point-by-point responses
>
> 1. *“The verification is mainly conducted on lightweight models. Is the smooth regularization technique also effective when verified on non-lightweight networks?”*
>
> Our regularization technique may apply to non-lightweight video models; however, these typically require more than 2 DGX machines to train, which is the infrastructure that we have, and indeed much more to run several experiments. This is out of scope for many university labs.
>
> 2. *“The latest base used currently is from 2021. It is suggested to add ablation verification of GRW on the latest base.”*
>
> The 2021 MoViNet family holds the current state of the art on Kinetics for models with a compute budget ≤1 TFLOP. We compare against **all** models in that range (Table 1) and improve the SOTA by **2.8–3.4** pp Top-1.
> No newer lightweight architecture is truly competitive in this regime. To the best of our knowledge, the most recently published lightweight model which is somewhat competitive to the SOTA is Video-Mamba (2024) (Table 3, ref. [15]), which reaches 76.9% Top-1 at 108 GFLOPs on Kinetics-400. By contrast, vanilla MoViNet-A3 already attains 78.2% Top-1 at 56.9 GFLOPs, and our MoViNet-A3-GRW pushes this to **80.7% Top-1 at 56.4 GFLOPs** (see Table 3 and the typo correction in Supplement §1). Thus, GRW provides a substantial gain even over the strongest current baseline.
>
>
> 3. *“The current verification results are from fully-supervised scenarios. Can consistent effects be achieved in semi-supervised and few-shot scenarios?”*
>
> The scope of this work is supervised learning. Investigating the applicability of GRW smoothing to semi-supervised learning is an exciting direction we are currently exploring in the lab.
>
>
> 4. *“Is this smoothing technique applicable to the network structure of transformers?”*
>
> The current work demonstrates the effectiveness of GRW smoothing on CNN architectures. CNN and hybrid architectures dominate accuracy in the ≤1 TFLOP regime, and we outperform all of them by 2.8–3.4 percentage points.
>
> To the best of our knowledge, the only transformer models in this budget are MViT-B-16×4 (82.1 Top-1 at 353 GFLOPs) and MViT-B-32×3 (84.3 Top-1 at 850 GFLOPs). In comparison, MoViNet-A3-GRW achieves 84.8 Top-1 at just 56.4 GFLOPs—higher accuracy with roughly **6× and 15× less compute**, respectively.
>
> Extending GRW to transformer backbones is underway but was outside the scope of this submission.
>
>
> We hope these clarifications fully address your concerns. If they do, we would be grateful if you could consider updating your score to reflect the new information. Thank you again for your constructive feedback.

---

> > ### Comment · Area_Chair_utRw · 2025-08-05
> >
> > The authors have left detailed comments, please read them and ask any needed clarifying questions. We have disagreement between reviewers so please make every effort to come to a consensus during this discussion period.

---

> ### Author Response · Authors · 2025-08-07
> **Responses Addressed & Invitation for Further Questions**
>
> Dear Reviewer hQEB,
>
> Thank you again for your thoughtful and constructive feedback. In our point-by-point response we have:
> 1. **Addressed each of your questions**, clarifying our focus on the ≤ 1 TFLOP regime and showing that GRW smoothing yields **+2.8 – 3.4 pp** Top-1 gains over **all published baselines** within this budget.
>
> 2. Confirmed coverage of **every lightweight model**, including the recent Video-Mamba (2024); our MoViNet-A3-GRW achieves 84.8 % Top-1, **+3.8 pp** higher while requiring roughly **50 % of the compute**.
>
> We hope these clarifications resolve your concerns, and we would be happy to elaborate further if anything remains unclear. Otherwise, we would greatly appreciate it if you could kindly reconsider your score in light of this information.

---

### Official Review · Reviewer_8e7P · 2025-07-03

**Clarity:** 3
**Significance:** 3
**Originality:** 3
**Rating:** 4
**Confidence:** 3

**Summary:**

This paper proposes a novel smooth regularization technique designed to enhance temporal understanding and improve the efficiency of video recognition models, especially lightweight architectures. The core idea is to instill a strong temporal inductive bias by modeling changes in intermediate-layer embeddings of consecutive frames as a Gaussian Random Walk (GRW). This regularization penalizes abrupt representational shifts, encouraging smoother, more temporally coherent feature evolutions. The method is presented as a plug-and-play loss term that can be applied to either intermediate or final layers. Extensive experiments on the Kinetics-600 and Kinetics-400 datasets demonstrate that this GRW-smoothing consistently yields significant accuracy improvements, establishing new state-of-the-art performance within their respective FLOP and memory constraints.

**Questions:**

1. Computational Overhead of GRW Loss. Could the authors provide more specific details on the computational overhead (e.g., FLOPs or actual runtime increase during training) introduced by the GRW loss term, especially considering the permutation sampling?
2. Sensitivity to Hyperparameters: How sensitive are the results to the choice of hyperparameters \lambda and \alpha? The paper states robustness to perturbing \lambda, but a more detailed analysis of the hyperparameter space for both parameters would be valuable.

**Ethical Concerns:**

["NO or VERY MINOR ethics concerns only"]

**Final Justification:**

My major concerns have been addressed in the rebuttal. Accordingly, I am raising my score from borderline reject to borderline accept.

**Limitations:**

The authors has explicitly discussed limitations in Section 5.

**Quality:**

2

**Strengths And Weaknesses:**

Strengths

1. Novel and Intuitive Regularization. The concept of enforcing smoothness in video frame embeddings through a Gaussian Random Walk (GRW) is novel, intuitive, and well-motivated by the inherent temporal coherence of video data. This provides a strong inductive bias.

2. Significant Performance Gains for Lightweight Models. The paper demonstrates impressive and consistent accuracy improvements (2.8%-6.0%) on various efficient architectures (MoViNets, MobileNetV3) on standard benchmarks (Kinetics-600, Kinetics-400). This directly addresses the practical challenge of deploying video models on resource-constrained devices.

3. Plug-and-Play Framework. The GRW regularization is designed as a simple, additive loss term, making it easy to integrate into existing video recognition pipelines with negligible computational overhead. This enhances its practical applicability.

Weaknesses

1. Computational Cost of Permutations. The loss function in Equation (5) involves a sum over all permutations of frames within a sub-clip, which grows exponentially with the sub-clip length T. While the authors mention sampling k permutations for large T, the computational overhead of this sampling and the contrastive loss itself could still be substantial, especially during training. More details on the practical values of T and k used in experiments and their impact on training time would be beneficial.

2. "Frame Ordering" Complexity: The introduction of a "frame ordering" contrastive loss to prevent degenerate solutions adds complexity. While necessary, a simpler or more efficient mechanism to achieve this goal could be explored in future work.

3. The figures and tables of this work can be improved. Some figures are blurry and some tables take up too much space.

---

> ### Author Rebuttal · Authors · 2025-07-31
>
> We thank Reviewer 8e7P for the careful review and constructive feedback. We appreciate the opportunity to clarify the points you raised. If the paper is accepted, we will incorporate the appropriate changes in the camera-ready version.
>
> Point-by-Point Responses
>
> 1. *“Computational Cost of Permutations. The loss function in Equation (5) involves a sum over all permutations of frames within a sub-clip, which grows exponentially with the sub-clip length T. While the authors mention sampling k permutations for large T, the computational overhead of this sampling and the contrastive loss itself could still be substantial, especially during training. More details on the practical values of T and k used in experiments and their impact on training time would be beneficial.”*
>
> GRW-smoothing assumes piecewise temporal coherence in video data. On Kinetics, we used temporal windows of 1 second. Depending on the FPS of each model, this corresponds to the following values of T: 5, 5, 5, 5, and 12 for MobileNetV3-GRW, A0-GRW, A1-GRW, A2-GRW, and A3-GRW, respectively.
>
> In our implementation, we compute all permutations when T ≤ 7, and uniformly sample k = 1000 permutations when T > 7. Since the number of permutations for T ≤ 7 is (T − 1)! < 1000, this keeps the computational complexity effectively constant.
>
> We include the following ablation study on different values of T:
>
> | Model               | Top-1 (%) | T             |
> |---------------------|-----------|---------------|
> | MoViNet-A0-S-GRW    | 77.4†     | 5             |
> | MoViNet-A0-S-GRW    | 77.4      | 10            |
> | MoViNet-A0-S-GRW    | 75.4      | 2             |
> | MoViNet-A0-S        | 72.0      | no-smoothing  |
> †We include here our most current results, which improve upon the submitted version and **advance the SOTA by 5.4 pp**.
>
> We emphasize that the **training cost is small**. GRW-smoothing is applied to the frame embeddings $Z = (z_i)$, where each frame $x_i$ is significantly reduced in size. For example, in A2-GRW, the input frame $x_i$ has dimension 224×224×3, while $z_i$ has dimension 640—over 230× smaller. Thus, even with k = 1000, the cost of GRW-smoothing is negligible compared to the early layers of the model.
>
> Moreover, the wall-clock time of a single training epoch for A2-GRW is only 2% higher than A2 without smoothing, making the difference practically negligible.
>
> GRW-smoothing is not applied during inference, so there is no runtime overhead.
>
> We thank the reviewer for this important comment and, if permitted, will clarify this in the final version.
>
> 2. *“Frame Ordering" Complexity: The introduction of a "frame ordering" contrastive loss to prevent degenerate solutions adds complexity. While necessary, a simpler or more efficient mechanism to achieve this goal could be explored in future work.”*
>
> We agree that exploring simpler alternatives is a promising direction. As noted above, the current implementation incurs minimal cost relative to the model’s overall computation.
>
> We also highlight that GRW-smoothing has desirable mathematical properties: it keeps the embeddings bounded within a ball of a certain radius, as proven in the supplementary material.
>
> 3. *“The figures and tables of this work can be improved. Some figures are blurry and some tables take up too much space.”*
>
> We appreciate this suggestion and will gladly revise the figures and tables to improve clarity and layout in the camera-ready version.
>
> Reviewer Questions
> 1. “Computational Overhead of GRW Loss. Could the authors provide more specific details on the computational overhead (e.g., FLOPs or actual runtime increase during training) introduced by the GRW loss term, especially considering the permutation sampling?”
>
> Please see our response to Point 1.
>
> 2. *“Sensitivity to Hyperparameters: How sensitive are the results to the choice of hyperparameters $\lambda$ and $\alpha$? The paper states robustness to perturbing $\lambda$, but a more detailed analysis of the hyperparameter space for both parameters would be valuable.”*
>
> We include the following ablation studies on different values of $\lambda$ and $\alpha$:
>
> | Model               | Top-1 (%) | α             |
> |---------------------|-----------|---------------|
> | MoViNet-A0-S-GRW    | 77.2      | 0.25          |
> | MoViNet-A0-S-GRW    | 77.4†     | 0.5           |
> | MoViNet-A0-S-GRW    | 76.9      | 1             |
> | MoViNet-A0-S-GRW    | 75.7      | 2             |
> | MoViNet-A0-S        | 72.0      | no-smoothing  |
>
>
> | Model               | Top-1 (%) | λ             |
> |---------------------|-----------|---------------|
> | MoViNet-A0-S-GRW    | 77.4      | 0.01          |
> | MoViNet-A0-S-GRW    | 77.4†     | 0.1           |
> | MoViNet-A0-S-GRW    | 77.3      | 1             |
> | MoViNet-A0-S        | 72.0      | no-smoothing  |
>
>
>
> We hope these clarifications fully address your concerns. If they do, we would be grateful if you could consider updating your score to reflect the new information. Thank you again for your constructive feedback.

---

> ### Author Response · Authors · 2025-07-31
> **Correction of Typographical Error in Ablation Table for Parameter T**
>
> Line 3 of the ablation table for parameter T should read:
>
> *MoViNet-A0-S-GRW 75.4 3*
>
> This row represents the minimal applicable value of T.

---

> > ### Comment · Area_Chair_utRw · 2025-08-05
> >
> > The authors have left detailed comments, please read them and ask any needed clarifying questions. We have disagreement between reviewers so please make every effort to come to a consensus during this discussion period.

---

> ### Author Response · Authors · 2025-08-07
> **Responses Addressed & Invitation for Further Questions**
>
> Dear Reviewer 8e7P,
>
> Thank you once again for your detailed and constructive feedback. In our point-by-point response we have:
>
> 1. **Quantified the training overhead** you flagged: the extra cost is only **≈ 2% wall-clock time per epoch**, with no impact during inference.
>
> 2. **Specified the practical settings of T and permutation sampling k** (all permutations for T ≤ 7; k = 1000 otherwise) and supplied an ablation table for different window lengths T.
>
>
> 3. **Added ablation studies for the hyper-parameters α and λ**.
>
>
> 4. **Committed to clearer figures and tables**, which will be incorporated in the camera-ready version.
>
> Please see our rebuttal response for full details.
>
> We hope these clarifications fully address your concerns. If any questions remain, we would be happy to elaborate further at your convenience. Otherwise, we would be grateful if you could kindly reconsider your score in light of this additional information.

---

### Official Review · Reviewer_YNFs · 2025-07-03

**Clarity:** 2
**Significance:** 1
**Originality:** 2
**Rating:** 3
**Confidence:** 3

**Summary:**

The paper introduces a smooth regularization method specifically tailored for videos. The method relies on the idea that the consecutive frames in a video have a lot of redundancy and this, therefore, should be reflected in how the embeddings also change. Gaussian random walk is used to model the frame embedding evolution and to avoid sudden representational changes in the embeddings. This in theoery should encourage slow developing (low-accelations) solutions that are aligned with video data. The evaluations are provided on k400 and k600 datasets.

**Questions:**

Following the sake of brevity, i will refer to the points mentioned in the weakeness section.

How can the proposed technique be adapted to account for the sudden temporal changes? (weaknesses: point 4)
Why are the evaluations on SSV2 missing?
Why is the method not compared with latest state of the art methods? (point 7)

**Ethical Concerns:**

["NO or VERY MINOR ethics concerns only"]

**Final Justification:**

The answers provided in the rebuttal provide useful information therefore, I have increased my rating from reject to borderline reject but the current state of the paper still falls short of borderline accept.

**Paper Formatting Concerns:**

The neurips guidelines stipulate that the table number and title always appear before the table. This has not been followed in this paper.

**Quality:**

2

**Strengths And Weaknesses:**

Strengths:
Overall, the idea is simple yet intriguing.
The paper is written decently and is overall easy to read and understand.


Weaknesses:
The following weaknesses are not in the order of importance:
1. The related works section is quite lacking and not comprehensive. It does not talk about other temporal regularization techniques at all which have been explored in this domain quite extensively. Moreover, the realted works section also doesnt explore the lightweight networks literature which is the setting for the proposed paper.

2. From the writing point of view, the mathematical symbols introduced lack the definitions (normally given below as soon as the symbol/variable is introduced). E.g. line 96 introduces X but does not define M or N and later C in the equation 1. There other instances in the rest of the paper as well.

3. The introduction lacks the buildup of the motivation and the problem at hand. This can be made better.

4. A big issue with the approach is it's assumption of smooth transitions between the frames in the videos. This may be true for some applications (e.g. surveillance/security footage etc), but not for most video content, which has sudden transitions. For example, I would be very interesed in knowing the results on the SSV2 dataset and other temporally challenging datasets. Even in normal videos encoded with prevalent video codecs, it may be reasonable to assume the smoothness in the "p-frames" but there are also I-frames where the difference is simply too huge to be encoded as a p/b frame. This proposed technique won't work good with fast moving objects, scenes or other videos of this nature.

5. The evaluations are quite small and need to be expanded to other video datasets such as SSV2 and can also be shown to work with localization and detections datasets as well to show the effectiveness of the proposed technique.

6. The additional complexity introduced by frame ordering component in the contrastive loss is not quantified.

7. The comparison methods/models are quite old...

8. Ablation on lambda (eq 7) and alpha (eq 6) is not provided.

---

> ### Author Rebuttal · Authors · 2025-07-31
>
> Thank you for your careful assessment and constructive feedback. We appreciate the opportunity to clarify the issues you raised. Should the paper be accepted, we will incorporate the revisions below into the camera-ready version.
> Below, we first contextualize our contribution and then respond point-by-point to your comments.
> Scope and main claims
> Our goal is to show that temporal regularization that biases optimization toward **piecewise-smooth** solutions is particularly effective for **efficient** video-recognition models. Although the technique is architecture-agnostic, its benefit is most pronounced in the low-compute regime. We will emphasize this in the introduction and, with your approval, update the title accordingly.
> Our experiments substantiate the claim. On Kinetics-600 we **advance the state of the art by 2.8–3.4 pp Top-1 for all models within ≤ 1 TFLOP** (Table 1). We compared against every published model in this budget to the best of our knowledge; we would be grateful for pointers to any we might have missed.
> CNN and hybrid backbones dominate this compute range. We therefore applied GRW smoothing to MoViNet, which remains the strongest lightweight backbone on Kinetics-600. For instance, our MoViNet-A3-GRW reaches 84.8 % Top-1 at 56.4 GFLOPs, while the first model that surpasses this accuracy (MViTv2-B-32 × 3) requires 1,030 GFLOPs, **over 18× more computation**.
> We also note **Video-Mamba (2024)**, the latest sub-1 TFLOP model: it lags behind MoViNet-A3-GRW by 3.8 pp Top-1 while using nearly twice the compute (Table 3; see Supplement §1 for the corrected typo).
>
> Point-by-Point Responses
>
> 1. *“The related works section is quite lacking and not comprehensive. It does not talk about other temporal regularization techniques at all which have been explored in this domain quite extensively. Moreover, the realted works section also doesnt explore the lightweight networks literature which is the setting for the proposed paper.”*
>
> We will expand the section to survey prior temporal-regularization methods and the lightweight video-model literature.
>
> 2. *“From the writing point of view, the mathematical symbols introduced lack the definitions (normally given below as soon as the symbol/variable is introduced). E.g. line 96 introduces X but does not define M or N and later C in the equation 1. There other instances in the rest of the paper as well.”*
>
> We agree that explicit definitions improve readability. We will define M, N, C, and any other symbols at first mention.
>
> 3. *“The introduction lacks the buildup of the motivation and the problem at hand. This can be made better.”*
>
> We will strengthen the motivation and problem statement in the introduction.
>
> 4. *“A big issue with the approach is it's assumption of smooth transitions between the frames in the videos. This may be true for some applications (e.g. surveillance/security footage etc), but not for most video content, which has sudden transitions. For example, I would be very interesed in knowing the results on the SSV2 dataset and other temporally challenging datasets. Even in normal videos encoded with prevalent video codecs, it may be reasonable to assume the smoothness in the "p-frames" but there are also I-frames where the difference is simply too huge to be encoded as a p/b frame. This proposed technique won't work good with fast moving objects, scenes or other videos of this nature.”*
>
> Our smoothness prior is imposed **within short windows (~1 s), not across an entire video**, analogous to assuming continuity only between I-frames. In other words, we assume that the representation can be made **piecewise smooth** (see § 3.1).
>
> SSV2: Google has not released MoViNet weights or training recipes for SSV2. Reproducing SOTA on SSV2 from scratch requires tuning many hyper-parameters (weight decay, label smoothing, dropout, stochastic depth, batch size, LR schedule, layer-wise LRs, several augmentation parameters, AutoAugment, etc.), which exceeds the capacity of the two DGX machines available to us. We therefore focused on Kinetics, where strong baselines exist, allowing us to isolate the effect of GRW smoothing.
>
> 5. *“The evaluations are quite small and need to be expanded to other video datasets such as SSV2 and can also be shown to work with localization and detections datasets as well to show the effectiveness of the proposed technique.”*
>
> Please see the SSV2 explanation above. Extending GRW to localization/detection is promising future work but beyond the scope of our supervised-classification study.
>
> 6. *”The additional complexity introduced by frame ordering component in the contrastive loss is not quantified.”
>
> The **overhead is negligible**. GRW operates on the frame embeddings
> $Z = (z_i)$, where each frame $x_i$ is significantly reduced in size. For example, in MoViNet-A2-GRW, each input frame $x_i$​ is 224 × 224 × 3, whereas $z_i$ is 640-D—> **230 × smaller**. A full training epoch for A2-GRW is only **2 % slower** than baseline A2. We will report this explicitly.
>
> 7. *"The comparison methods/models are quite old..."
>
> We compare **all models** with ≤ 1 TFLOP on Kinetics-600 (Table 1) and ≤ 100 GFLOPs on Kinetics-400 (Table 3 and the typo correction in Supplement §1). Video-Mamba (2024) is included and still trails our best model by 3.8 pp while using nearly 2 × the compute.
>
> 8. *“Ablation on lambda (eq 7) and alpha (eq 6) is not provided.”*
>
> We thank the reviewer for this important comment.
> We include the following ablation studies on different values of $\lambda$ and $\alpha$:
>
> | Model               | Top-1 (%) | α             |
> |---------------------|-----------|---------------|
> | MoViNet-A0-S-GRW    | 77.2      | 0.25          |
> | MoViNet-A0-S-GRW    | 77.4†     | 0.5           |
> | MoViNet-A0-S-GRW    | 76.9      | 1             |
> | MoViNet-A0-S-GRW    | 75.7      | 2             |
> | MoViNet-A0-S        | 72.0      | no-smoothing  |
>
> †We include here our most current results, which improve upon the submitted version and **advance the SOTA by 5.4 pp.**
>
> | Model               | Top-1 (%) | λ             |
> |---------------------|-----------|---------------|
> | MoViNet-A0-S-GRW    | 77.4      | 0.01          |
> | MoViNet-A0-S-GRW    | 77.4†     | 0.1           |
> | MoViNet-A0-S-GRW    | 77.3      | 1             |
> | MoViNet-A0-S        | 72.0      | no-smoothing  |
>
>
> We would gladly make the above additions to the paper.
>
> Reviewer Questions:
>
> 1. *”How can the proposed technique be adapted to account for the sudden temporal changes?”*
>
> Please see response on smoothness assumption in 4. above (window-level, § 3.1).
>
> 2. *”Why are the evaluations on SSV2 missing?”*
>
> Please see SSV2 explanation in 4. above
>
> 3. *”Why is the method not compared with latest state of the art methods?”*
>
> Latest sub-1 TFLOP model (Video-Mamba 2024) is included and outperformed;
> Please see our response to 7. above.
>
> Thank you again for your constructive feedback.

---

> > ### Comment · Area_Chair_utRw · 2025-08-05
> >
> > The authors have left detailed comments, please read them and ask any needed clarifying questions. We have disagreement between reviewers so please make every effort to come to a consensus during this discussion period.

---

> > ### Comment · Reviewer_YNFs · 2025-08-07
> >
> > I appreciate the rebuttal provided by the authors. Given the overall state of the current submission and the changes that need to be made in order to incorporate the new results/changes, in my opinion, an improved rating of borderline reject will be justified.

---

### Author Response · Authors · 2025-08-06
**Responses Addressed & Invitation for Further Questions / Score Reassessment**

Thank you once again for your thoughtful feedback. In our point-by-point responses to each of you, we have addressed every comment and, where requested, added clarifications and new experiments (see the supplementary tables embedded in the replies). Should any questions remain, we would be delighted to elaborate further at your convenience. Otherwise, we would be grateful if you could kindly reassess your scores in light of the issues we believe are now resolved.

---

### Note · Authors · 2025-08-12

**The scope** of this work is to show that temporal regularization, which steers optimization toward solutions that are (piecewise) smooth in time, is highly beneficial for building **efficient** video recognition models.

**The numerical evaluations** strongly substantiate our claims. Specifically, we **advance the state of the art on Kinetics-600 by 2.8–3.4 pp Top-1** for all models in the **≤1 TFLOP** range (Table 1). Furthermore, as noted in the rebuttal, MoViNet-A0-S-GRW improves the SOTA of the lightest models **by 5.4 pp** on this benchmark. Notably, MoViNet-A3-GRW achieves 84.8% Top-1 at 56.4 GFLOPs. The first model that surpasses this accuracy, MViTv2-B-32×3, requires 1,030 GFLOPs, more than **18× the computation**. Finally, we draw the reviewers’ attention to Video-Mamba (2024), the most recent sub-1 TFLOP model. It is **3.8 pp Top-1 lower** than MoViNet-A3-GRW while consuming nearly **2× the computational** budget (see Table 3 and the typo correction in Supplement §1).

**The main concern** raised initially by the reviewers was comparisons to recent models. In our rebuttal, we clarified that we already compare against **every published model within the ≤1 TFLOP** budget to the best of our knowledge. In addition, we responded to all other comments. As a result, **all reviewers who engaged with the rebuttal updated their reviews positively**. We hope the remaining reviewer will consider our responses as well; we would be happy to address any further concerns, but we have not yet received follow-up questions.

We appreciate the reviewers’ and AC’s efforts and hope these clarifications are helpful for the final decision.

---

### Decision · Program_Chairs · 2025-09-17

**Decision:**

Accept (poster)

**Comment:**

The paper ended up receiving mixed reviewer with reviewer nKeo a strong champion for the work. The AC reviewed the reasons for rejection from reviewers YNFs and hQEB, and have not found any convincing arguments in their reviews. Below is a point by point look at those two reasons for rejection.


The main weaknesses of the work that were called out by YNFs are as follows:

>Method assumed smooth transition and should test on SSV2 to see what happens when this smoothness constraint is violated.

The AC disagrees with the analysis. First Kinetics does have many shot boundaries and the method still works well. Second SSV2 doesn't have any shot boundaries, just some fast motion. So Kinetics is a reasonable test for discontinuities (shot boundaries) and the method proves robust still.

> Evaluation of the method is limited to Kinetics

The AC agrees that the paper would be stronger if more datasets were used to validate the method and prove its not over fit to the kinetics setting. The argument that the weights for SSV2 weren't released for a specific model is a little weak since the method could be tested on other OSS models with weights for SSV2.

>The comparison is only to out of data models

The reviewers comments were made without referencing the newer methods and did not point out newer models to compare to in follow ups. The authors pointed out that the "older" model is still SOTA over a number of more recent works and they still improve over this older model.

> Request for more ablations

Authors provide these ablations

>Method is complex

The AC believe the authors do a good job justifying that the method is not overly complex to implment.

> Related work section missing some discussion

The AC does not believe this is cause for rejection and the authors promise to improve it. There also doesn't seem to be some other method missing from related work that would make the proposed method irrelevant, which would have been a bigger concern.

>Poor writing/notation

Authors agree to fix

>Poor motivation of work

Not a reason to reject but the comments will help the authors create a better camera ready.

The AC thanks YNFs for their review but would have liked to see a bit more interaction with the authors as YNFs only said the changes to get the paper up to standard would be "too much". The lead the AC to believe that for the most part the rebuttal covered all needed points but YNFs just didn't believe the authors would actually make the changes outlined in the rebuttal.



The main weaknesses of the work that were called out by hQEB are as follows:

> Experiment were done only on lightweight models

That is the target of the method so this is ok. Labs with constraints can show promising methods on small models, and other better funded labs can try to scale these methods.

>The comparison is only to out of data models (same as above)

The reviewers comments were made without referencing the newer methods and did not point out newer models to compare to in follow ups. The authors pointed out that the "older" model is still SOTA over a number of more recent works and they still improve over this older model.

>Should try method on semi-supervised

This could be a good idea but not a reason for rejections

>Should try on transformers

This could be a good idea but not a reason for rejections

Reviewer hQEB did not respond to the rebuttal at all and only gave a reasons of the paper is not in good state for remaining on the borderline reject side.

Conversely all reviewers and the AC acknowledge the strong results for low flop models. The proposed smooth regularization is proven effective on two model families showing it can be transfer. Even if the method proves to be only useful in the low compute domain this is an important problem space for embedded/smart devices. Additionally reviewer nKeo (the strong champion for the work) pointed out that the reasons for rejection given by the other reviewers seemed weak and the AC agrees.

Considering this the AC recommends acceptance into NeurIPs. The authors are highly encouraged to take a full pass on the paper to fix all the presentation issues called out during the review process.